# ALPHARoPE: A SIMPLE YET EFFECTIVE LENGTH EXTRAPOLATION METHOD

## ABSTRACT

Long-context modeling has become the core capability for Large Language Models (LLMs) and most of the studies focus on scaling Rotary Position Embeddings (RoPE) to overcome the inherent limitations of positional encoding extrapolation. Previous methods show that the high-frequency components of RoPE are sensitive to small relative distances and capture local information, while the low-frequency components respond to large relative distances and capture long-range dependencies. This phenomenon has led to the conventional strategies of directly extrapolating the high-frequency components and interpolating the low-frequency ones. However, due to the periodic nature of trigonometric functions, appropriate interpolation of high-frequency components can enhance their ability to capture longer-range dependencies, thereby contributing to improved long-context modeling. Building on these insights, we propose AlphaRoPE, a novel approach for RoPE-based length extrapolation. AlphaRoPE applies interpolation to low-frequency components to resolve out-of-distribution (OOD) issues, while for the high-frequency components, it introduces a carefully calibrated, gradually increasing interpolation factor as frequency descends. This dual approach effectively extends the context window of LLMs without degrading their performance on shorter sequences. Experiments conducted on various models further confirm our hypothesis and demonstrate the superiority of AlphaRoPE.

## 1 INTRODUCTION

In recent years, the ability to handle long-context tasks has become a crucial evaluation criterion for Large Language Models (LLMs). With growing application demands, such as full-document analysis (Beltagy et al., 2020), long dialogue modeling (Guo et al., 2021), and complex reasoning (Wei et al., 2022), the ability to capture long-range dependencies has gained significant attention. Leading models like GPT-5 (OpenAI, 2024), DeepSeek-R1 (Guo et al., 2025), Gemini2.5 (Comanici et al., 2025), Claude4 (Anthropic, 2024), and Llama3.1 (Dubey et al., 2024) now support substantially extended context windows, establishing long-context modeling capability as a standard feature in next-generation LLMs.

The key factor for this context expansion lies in adapting positional encoding mechanisms (Vaswani et al., 2017) during the continual pre-training stage. Rotary Position Embedding (RoPE) (Su et al., 2024) is widely adopted due to its favorable extrapolation properties. A common strategy involves rescaling RoPE to accommodate longer sequences and then fine-tuning on long-context data to ensure consistent positional awareness. Several methods have been proposed to determine suitable scaling strategies for RoPE-based length extrapolation, including YaRN (Peng et al., 2023), NTK (LocalLLaMA, 2023), and LongRoPE (Ding et al., 2024). They adopt rescaling factors that remap extended token positions into the value range that the model learned to handle during pre-training, thus solving the out-of-distribution (OOD) problem in RoPE. However, these studies often employ a piecewise scaling approach, sacrificing fidelity for simplicity. YaRN, for instance, operates under a "high-frequency extrapolation, low-frequency interpolation" paradigm, assuming that high-frequency dimensions are only relevant for short-range dependencies and should remain unscaled to preserve local information. We find that due to the inherent periodicity of trigonometric functions, high-frequency dimensions can also contribute to encoding long-range dependencies. Completely omitting them from the interpolation process is an oversimplified design that limits their potential.

Our method is motivated by the insight that a controlled, small-amplitude interpolation of high-frequency dimensions can better adapt the model to longer sequences without compromising its ability to capture local context. A critical issue with existing methods is that their scaling factors on non-OOD dimensions are excessively large, leading to unnecessary positional disturbances. This is particularly problematic for high-frequency dimensions, which are already well-trained during the pre-training phase to model short-range dependencies. Ideally, the scaling for these dimensions should be minimal to avoid disrupting the learned representations and to reduce the number of fine-tuning tokens required for adaptation.

To achieve this, we introduce AlphaRoPE, a new RoPE scaling strategy based on a mathematically-derived power function that elegantly satisfies our design principles. We further present a novel metric, the high-frequency scaling magnitude $A$, to quantify the degree of interpolation on these critical non-OOD dimensions. Our analysis demonstrates that as the context window expands, the value $A$ of our method grows at a significantly lower rate than existing methods, providing theoretical evidence of our method's superior extensibility and training efficiency. Experimental results validate our approach, with AlphaRoPE achieving state-of-the-art performance on diverse long-context tasks. Its effectiveness on various models not only demonstrates superior performance over strong baselines like YaRN but also proves its generalizability.

Our main contributions are summarized as follows: 1) Theoretical Re-examination: We challenge the conventional wisdom of "high-frequency extrapolation" by demonstrating that appropriate interpolation of high-frequency RoPE dimensions can enhance their contribution to long-range dependency encoding through trigonometric periodicity, providing new insights for positional encoding design. 2) Novel Scaling Framework: We propose AlphaRoPE, a mathematically-derived scaling strategy that achieves optimal balance between local preservation and long-context adaptation through controlled high-frequency interpolation. 3) Comprehensive Validation: We introduce a quantitative metric for scaling magnitude and demonstrate state-of-the-art performance on diverse long-context benchmarks across multiple model architectures.

# 2 Background and Related Work

## 2.1 Rotary Position Embedding (RoPE)

Our work is based on the Rotary Positional Embedding (Su et al., 2024), which becomes the de facto module in modern LLMs. Let $m, n$ be the positional index of $\mathbf{x}_m, \mathbf{x}_n \in \mathbb{R}^d$, where $d$ denotes the attention-head dimension. RoPE converts them into query and key vectors:

$$\mathbf{q}_m = f_q(\mathbf{x}_m, m), \quad \mathbf{k}_n = f_k(\mathbf{x}_n, n) \tag{1}$$

where $f_q$ and $f_k$ are the transformation functions for queries and keys, respectively. The core idea of RoPE is to encode absolute positional information with a rotation matrix. For a vector $\mathbf{x} \in \mathbb{R}^2$ representing a pair of features, the rotation matrix is defined as:

$$\mathbf{R}_{\Theta,m} = \begin{pmatrix} \cos m\theta & -\sin m\theta \\ \sin m\theta & \cos m\theta \end{pmatrix} \tag{2}$$

For high-dimensional vectors in $\mathbb{R}^d$ (where $d$ is even), the rotation is applied to each pair of consecutive dimensions. The full block-diagonal rotation matrix $\mathcal{R}_{\Theta,m} \in \mathbb{R}^{d \times d}$ is:

$$\mathcal{R}_{\Theta,m} = \mathrm{diag}\left(\mathbf{R}_{\Theta_0,m}, \mathbf{R}_{\Theta_1,m}, \ldots, \mathbf{R}_{\Theta_{d/2-1},m}\right) \tag{3}$$

where $\Theta = \{\theta_i = b^{-2j/d}, j = 0, 1, 2, \ldots, d/2 - 1\}$ are the frequencies. The query and key vectors are then transformed as:

$$f_q(\mathbf{x}_m, m) = \mathcal{R}_{\Theta,m}\mathbf{W}_q\mathbf{x}_m, \quad f_k(\mathbf{x}_n, n) = \mathcal{R}_{\Theta,n}\mathbf{W}_k\mathbf{x}_n \tag{4}$$

where $\mathbf{W}_q$ and $\mathbf{W}_k$ are learned projection matrices. This rotational encoding ensures that the inner product between query and key vectors depends only on their relative position $m - n$:

$$(\mathcal{R}_{\Theta,m}\mathbf{q})^\top (\mathcal{R}_{\Theta,n}\mathbf{k}) = \mathbf{q}^\top \mathcal{R}_{\Theta,n-m}\mathbf{k} \tag{5}$$

thereby providing relative positional information in the attention mechanism.

## 2.2 RoPE Scaling

When the target sequence length exceeds the pre-trained window length, LLMs cannot directly extrapolate, as they encounter previously unseen positional indices. This leads to an Out-of-Distribution (OOD) issue. Specifically, for the high-dimensional part of RoPE, the model does not complete a full period during pre-training. Assuming the base of RoPE is $b$ and the per-head dimension is $d$, and the model's pre-trained text window length is $L_{train}$, if $L_{train}b^{-2j/d} < 2\pi$ is satisfied, the model has not completed a full period beyond the dimension $2j$. This means that a portion of the angles are outside the model's training range. Consequently, when the target length $L_{target}$ exceeds $L_{train}$, angles outside the training range will inevitably appear beyond the dimension $2j$, which is the root cause of the OOD problem.

To address the OOD issue, the concept of a critical dimension is proposed. We follow the notation from (Liu et al., 2023) and denote the critical dimension, where $L_{train}b^{-2j/d} = 2\pi$, as $d_0$. This gives us:

$$d_0 = 2\lfloor \frac{d}{2} \log_b \frac{L_{train}}{2\pi} \rfloor \tag{6}$$

It is evident from the formula that for high dimensions where $j > d_0$, the condition $L_{train}b^{-2j/d} < 2\pi$ holds true. This implies that a portion of the angles for these dimensions remains untrained. Therefore, these high-dimensional parts require interpolation. We scale the out-of-training-range angles by an interpolation factor, $s = \frac{L_{target}}{L_{train}}$, to ensure all angles fall within the model's trained distribution.

Conversely, for the low dimensions where $j < d_0$, we have $L_{train}b^{-2j/d} > 2\pi$. This means these dimensions have completed at least one full rotational period during training. The smaller the value of $j$, the more periods are completed, indicating more thorough training for these dimensions.

## 2.3 Related Work

Several prominent methods have been developed to address the challenge of extending the context window of RoPE-based models. These approaches can be broadly categorized into positional interpolation and base modification techniques.

**PI** (Chen et al., 2023) scaling method introduces linear positional interpolation, applying a uniform scaling factor $s = \frac{L_{target}}{L_{train}}$ across all RoPE dimensions. This uniform scaling, however, leads to an issue of "overly crowded" positional information, where the entire encoding space is compressed. Consequently, the model's ability to distinguish between different positions is impaired due to the loss of discriminative resolution in the interpolated embeddings.

**NTK** (LocalLLaMA, 2023) scaling methods enhance RoPE-based models' ability to extrapolate by increasing the original RoPE base value to a larger value. An early method, proposed by (LocalLLaMA, 2023), enlarged the base by a factor of $s^{\frac{d}{d-2}}$. However, this approach was not aligned with OOD theory for high RoPE dimensions, resulting in insufficient interpolation and a drop in performance at longer sequence lengths.

A more effective and widely adopted NTK-based solution, introduced by (Liu et al., 2023), enlarges the RoPE base based on the theoretical critical dimension, $d_0$. This approach specifically enlarges the RoPE base by a factor of $s^{\frac{d}{d_0}}$, which provides a scaling factor $s_i \geq \frac{L}{L_{train}}$ for dimensions $i > d_0$.

Despite its effectiveness, this configuration can lead to excessive interpolation of the low-frequency components where $j > d_0$, requiring a significant increase in the number of tokens needed for fine-tuning (Liu et al., 2023). To address this, a **uniform interpolation factor** of $s$ is adopted for dimensions beyond the critical dimension. In this work, the term "NTK scaling" refers to this refined approach.

**YaRN** (Peng et al., 2023) significantly improves length extrapolation performance by treating different RoPE dimensions based on their frequency. It divides the dimensions into a high-frequency part and a low-frequency part. The high-frequency part, which corresponds to the low-dimension part of

RoPE, is handled with direct extrapolation. Conversely, the low-frequency part, which corresponds to the high-dimension part, undergoes linear interpolation to mitigate the OOD problem.

**LongRoPE** (Ding et al., 2024) distinguishes itself from theoretical derivation-based approaches by utilizing a performance-driven evolutionary search to optimize dimension-specific scaling parameters. Since this method requires additional computational costs and does not provide explicit scaling rules, we do not include it in our comparison and analysis.

# 3 AlphaRoPE

## 3.1 Rethinking High-Frequency Interpolation: Beyond Local, Toward Global Context

Current RoPE scaling methods, such as YaRN, predominantly follow a "high-frequency extrapolation, low-frequency interpolation" paradigm. This approach is based on the assumption that high-frequency dimensions (corresponding to small $j$ values) are primarily responsible for encoding local positional information, and should therefore be excluded from interpolation to preserve fine-grained positional accuracy.

We argue that this conventional view requires reconsideration. The fundamental mathematical form of RoPE encoding for dimension $j$ at position $m$ is:

$$\text{PE}(m, j) = e^{im\theta_j}, \quad \theta_j = b^{-2j/d} \tag{7}$$

This formulation exhibits inherent periodicity with period $T_j = 2\pi/\theta_j$, satisfying:

$$\text{PE}(m + kT_j, j) = \text{PE}(m, j) \quad \text{for any integer } k \tag{8}$$

This periodic property means that high-frequency dimensions (small $j$, small $T_j$) can encode information about positions separated by integer multiples of their period through aliasing effects, contrary to the assumption that they are solely dedicated to local information encoding.

The critical trade-off in RoPE scaling can be understood through the positional discrimination capability, which is quantified by:

$$|\text{PE}(m + \Delta, j) - \text{PE}(m, j)| = |e^{im\theta_j}(e^{i(\Delta \cdot \theta_j)} - 1)| = |e^{i(\Delta \cdot \theta_j)} - 1| = 2\left|\sin\left(\frac{\Delta \cdot \theta_j}{2}\right)\right| \tag{9}$$

As illustrated in Figure 1, this leads to two distinct operational regimes: High-frequency dimensions (small $j$) exhibit strong discrimination for small $\Delta$ (high local sensitivity) but suffer from frequent aliasing due to short $T_j$. Low-frequency dimensions (large $j$) provide long aliasing distance (large $T_j$) but show weak discrimination for adjacent positions.

Interpolation through scaling $\theta'_j = \theta_j/s$ extends the effective period to $T'_j = s \cdot T_j$, thereby increasing the aliasing distance. However, this operation simultaneously reduces local discrimination capability according to the sensitivity equation above.

Therefore, completely avoiding high-frequency interpolation (as in YaRN) sacrifices long-range potential, while excessive interpolation (as in NTK) harms local information preservation. This trade-off is evident in specific dimensions: interpolating the highest-frequency dimension (period $\approx$6.28) severely impairs local sensitivity with minimal long-range gain (period $\times$2 $\approx$12.56), whereas interpolating medium-frequency dimensions (e.g., period $\approx$112) meaningfully extends the effective range (period $\times$2 $\approx$224) with limited sensitivity loss.

Our method, AlphaRoPE, introduces a frequency-adaptive strategy that balances this trade-off: applying minimal interpolation to highest-frequency dimensions to preserve local sensitivity, while progressively increasing interpolation strength for lower frequencies to enhance long-context capability.

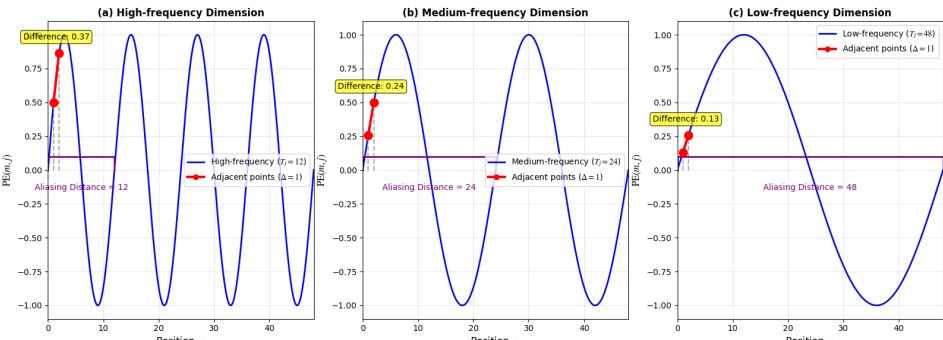

Figure 1: The trade-off between local sensitivity and aliasing distance in RoPE dimensions. (a) A high-frequency dimension (short period $T_j = 12$) exhibits large differences between adjacent positions ($\Delta = 1$), indicating high local sensitivity, but a short aliasing distance where positions separated by $T_j$ share the same encoding. (b) A medium-frequency dimension (period $T_j = 24$) achieves a compromise between local sensitivity and aliasing distance, maintaining reasonable positional discrimination while extending the effective range. (c) A low-frequency dimension (long period $T_j = 48$) has a long aliasing distance, supporting better long-range discrimination, but shows minimal differences between adjacent positions, resulting in low local sensitivity. This illustrates the fundamental compromise that guides our interpolation strategy.

## 3.2 Scaling Principles

To achieve this, we establish four fundamental design principles for scaling the RoPE dimensions. An ideal interpolation factor $f$ should satisfy:

1. **Monotonicity**: The interpolation factor $f(j)$ should increase monotonically with the dimension $j$, meaning that higher-frequency dimensions (smaller $j$) should undergo smaller interpolation magnitudes.

2. **Boundary Conditions**: At dimension $j = 0$, the interpolation factor $f(0) = 1$; at the critical dimension $d_0$, the interpolation factor $f(d_0) = s$, where $s$ is the overall scaling factor.

3. **Minimal High-Frequency Interpolation**: For smaller $j$ (i.e., high-frequency dimensions), the interpolation factor should be close to 1, ensuring minimal change in the interpolation magnitude for high-frequency dimensions. This preserves the model's ability to model local relationships and minimizes the difficulty of fine-tuning on short texts.

4. **Uniform Low-Frequency Interpolation**: According to 2.2, for all $j > d_0$ RoPE dimensions, which are all OOD dimensions, the interpolation factor should be **no less than** $s = \frac{L_{target}}{L_{train}}$.

The first three principles are applied to dimensions where $j \leq d_0$, while only the fourth principle is applied to dimensions where $j > d_0$. Following the YaRN setup, we can simply set $f(j) = s$ for $j > d_0$ to satisfy the fourth principle. To satisfy the first three principles, we naturally consider using a power function as our interpolation factor function for dimensions where $j \leq d_0$. Given the nature of a power function, the setup of $f(j) = s^{(j/d_0)^{\alpha}}$ for $j \leq d_0$ with $\alpha > 0$ perfectly satisfies the first three conditions.

Finally, we propose AlphaRoPE scaling:

$$f(j) = \begin{cases} s^{(j/d_0)^{\alpha}} & j \leq d_0 \\ s & j > d_0 \end{cases} \tag{10}$$

Notably, when $\alpha = 0$, the scaling factor $f(j)$ is uniformly $s$, which degenerates to PI interpolation. When $\alpha = 1$, the method becomes equivalent to NTK scaling. Therefore, by setting $\alpha > 1$, we can ensure that the interpolation magnitude of AlphaRoPE is smaller than that of NTK scaling.

## 3.3 Parameter Optimization and Metric Quantification

After defining our function form $f(j) = s^{(j/d_0)^{\alpha}}, j \leq d_0$, a key question is how to choose the parameter $\alpha$. For the high-frequency part, since there is no OOD issue, we aim to keep the interpolation magnitude as small as possible to reduce the number of fine-tuning steps required for the model. Specifically, we want $\alpha$ to be a function of $s$, ensuring that the degree of interpolation for high frequencies does not increase too rapidly with the scaling factor $s$.

We define a metric, $A$, to quantify the interpolation magnitude of high-frequency dimensions. And we aim to derive the value of $\alpha$ using this metric. $A$ metric is defined as the geometric mean of the interpolation factors for these dimensions:

$$A = \left( \prod_{j=1}^{d_0/2} s^{\left( \frac{2j}{d_0} \right)^{\alpha}} \right)^{\frac{2}{d_0}} = s^{\frac{2}{d_0} \sum_{j=1}^{d_0/2} \left( \frac{2j}{d_0} \right)^{\alpha}} \tag{11}$$

Next, we need to estimate $\frac{2}{d_0} \sum_{j=1}^{d_0/2} \left( \frac{2j}{d_0} \right)^{\alpha}$. Since $d_0 = 2 \lfloor \frac{d}{2} \log_{base} \frac{L_{train}}{2\pi} \rfloor$ is guaranteed to be even, $\frac{d_0}{2} = n$ is an integer. Thus, the expression becomes:

$$\frac{2}{d_0} \sum_{j=1}^{n} \left( \frac{j}{n} \right)^{\alpha} = \frac{\sum_{j=1}^{n} j^{\alpha}}{n^{\alpha+1}} \approx \frac{\int_0^n x^{\alpha} \, dx}{n^{\alpha+1}} = \frac{\frac{n^{\alpha+1}}{\alpha+1}}{n^{\alpha+1}} = \frac{1}{\alpha+1} \tag{12}$$

This approximation is reasonably accurate. Substituting into Eq. 11 yields:

$$A = s^{\frac{2}{d_0} \sum_{j=1}^{d_0/2} \left( \frac{2j}{d_0} \right)^{\alpha}} \approx s^{\frac{1}{\alpha+1}} = \exp \left( \frac{\ln s}{\alpha+1} \right) \tag{13}$$

The final metric $A$ is independent of the critical dimension and depends solely on the scaling factor $s$ and the parameter $\alpha$. To suppress the growth rate of the interpolation magnitude as $s$ increases, we define $\alpha$ as a function of $s$. We adopt a logarithmic form, $\alpha = 0.6 \ln s$, which ensures a slow growth rate and an upper bound $\frac{1}{0.6}$ on the key term. The derivation and empirical justification for the coefficient value of 0.6 are provided in Appendix C. This particular choice of constant guarantees that for common scaling factors ($s \geq 8$), we have $\alpha > 1$, which makes the interpolation amplitude smaller than that of the NTK method ($\alpha = 1$). For the special case where $s \leq 4$, this function yields $\alpha < 1$; therefore, we stipulate $\alpha = 1$ to maintain consistency with NTK scaling. This piecewise definition ensures a suppressed interpolation rate while aligning with NTK scaling for small factors.

Finally, we present the expression of AlphaRoPE:

$$f(j) = \begin{cases} s^{(j/d_0)^{\max(0.6 \ln s, 1)}} & j \leq d_0 \\ s & j > d_0 \end{cases} \tag{14}$$

## 3.4 Bridging the Spectrum: A Unified Analysis of RoPE Scaling Strategies

Following (Shang et al., 2025), we visualize AlphaRoPE's interpolation scaling factor, as shown in Figure 2, occupies a middle ground between NTK-aware scaling and YaRN. Its curve demonstrates a more gradual decay at lower frequencies compared to NTK, while maintaining a higher scaling factor than YaRN at higher frequencies. This design positions AlphaRoPE as a compromise between the two methods, balancing the need for controlled interpolation across all dimensions.

To quantitatively analyze the differences in interpolation amplitudes among various RoPE scaling methods, we examine how the value $A$—representing the average interpolation magnitude of high-frequency dimensions—varies with the scaling factor $s$ for each method. For the PI method, the interpolation factor is uniformly $s$ across all RoPE dimensions, making its value $A$ consistently $s$ and independent of the specific model. The NTK scaling method is a special case with $\alpha = 1$, and its value $A$ is derived following the same calculation framework. For the YaRN method, the value $A$ is

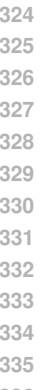

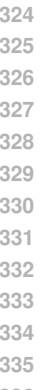

Figure 2: Per-dimensional RoPE scaling factor under different methods (Llama2-7B, $s = 16$)

contingent on the model's critical dimension, $d_0$, which is model-dependent. We use the Llama2-7B model as a concrete example, with its critical dimension of $d_0 = 90$, to calculate the value $A$ for YaRN. Further details on this calculation are provided in Appendix D.

| Method | $s = 8$ | $s = 16$ | $s = 32$ | $s = 64$ |
|--------|---------|----------|----------|----------|
| PI | 8 | 16 | 32 | 64 |
| NTK | 2.89 | 4.12 | 5.88 | 8.38 |
| YaRN | 1.99 | 2.32 | 2.61 | 2.85 |
| AlphaRoPE | 2.58 | 2.92 | 3.20 | 3.44 |

Table 1: A-value comparison under different scaling factors $s$ on Llama2-7b model

Table 1 presents the A values for various RoPE scaling methods under different scaling factors ($s = 8, 16, 32, 64$). Based on the value $A$ comparison across different scaling factors, several key observations emerge. First, NTK exhibits rapid growth in values $A$, while both YaRN and Alpha-RoPE demonstrate much more gradual increases, indicating better stability. Second, Alpha-RoPE consistently shows higher values $A$ than YaRN across all scales, confirming that its mild interpolation strategy in high-frequency components provides better mathematical properties compared to YaRN's approach of no interpolation in very high frequencies.

The comparison with NTK scaling, which corresponds to the special case of AlphaRoPE with $\alpha = 1$ serves as a baseline to validate the effectiveness of using the optimized $\alpha = 0.6 \ln s$. This optimized parameter significantly slows the growth of the value $A$, demonstrating an improvement over the NTK approach. Similarly, the systematic comparison with YaRN highlights the benefit of applying a small degree of interpolation to the high-frequency dimensions rather than completely avoiding it.

# 4 Experiments

## 4.1 Training Setup

We selected base models with relatively small original context windows to better evaluate length extension capabilities, choosing Llama2-7B (Touvron et al., 2023) (4k context) and OLMo-7B (Groeneveld et al., 2024) (2k context) for our experiments. This contrasts with models like Llama3.1 (Dubey et al., 2024), which already natively supports 128k context length. We extended these models to 64k context length, corresponding to scaling factors of $s = 16$ for Llama2-7B and $s = 32$ for OLMo-7B respectively. Inspired by CodeLlama (Roziere et al., 2023), which is trained on a 16k dataset with $s \approx 88.6$, we trained the Llama2-7B model with 16k context length datasets (approximately 0.6B tokens). We used the PG-19 dataset (Rae et al., 2019) chunked into 16k segments and trained for 600 steps. For OLMo-7B, we found that training on the 16k dataset did not yield satisfac-

tory results, so we used the PG-19 dataset (Rae et al., 2019) chunked into 32k segments and trained for 400 steps (approximately 0.8B tokens). For all model training experiments, We used the AdamW (Kingma & Ba, 2014) optimizer with a learning rate of $2 \times 10^{-5}$, $\beta_1 = 0.9$, and $\beta_2 = 0.95$. Weight decay was not applied, and we included a linear warmup for the first 20 steps. Our experimental setup roughly followed (Peng et al., 2023).

## 4.2 Evaluation

Our evaluation methodology is designed to assess both long and short-context performance. We evaluate long-context capabilities through three key scenarios: perplexity scores, the passkey retrieval task (Kamradt, 2023), and real-world long-context tasks from LongBench (Bai et al., 2023). For completeness, we also verify the model's performance on standard short-context benchmarks, and the corresponding results are provided in Appendix E.

### 4.2.1 Perplexity Score

| Model Name | 2k | 4k | 8k | 16k | 32k | 48k | 64k |
|---|---|---|---|---|---|---|---|
| Llama2-NTK ($s = 16$) | 4.387 | **3.827** | 3.345 | 2.850 | 2.597 | 2.531 | 2.519 |
| Llama2-YaRN ($s = 16$) | 4.360 | 3.845 | 3.351 | 2.887 | 2.662 | 2.603 | 2.589 |
| Llama2-AlphaRoPE ($s = 16$) | **4.351** | 3.838 | **3.324** | **2.837** | **2.593** | **2.529** | **2.508** |
| OLMo-NTK ($s = 32$) | 5.931 | 4.948 | 4.317 | 3.635 | 3.280 | 3.190 | 3.124 |
| OLMo-YaRN ($s = 32$) | 5.872 | 5.003 | 4.392 | 3.731 | 3.382 | 3.298 | 3.213 |
| OLMo-AlphaRoPE ($s = 32$) | **5.767** | **4.874** | **4.252** | **3.586** | **3.236** | **3.148** | **3.076** |

Table 2: Sliding Window Perlexity (S = 128) on Proof-pile documents over extended Llama2-7B and OLMo-7B.

Following the setup of (Peng et al., 2023), we use the Proof-Pile (Azerbayev et al., 2022) dataset to evaluate the perplexity scores of the extended Llama2-7B model and OLMo-7B model. We use the sliding window method from (Press et al., 2021) with $S = 128$.

Table 2 compares the performance of Llama2-7B and OLMo-7B models when their context windows are extended to 64k from 4k and 2k, respectively. The results show that our AlphaRoPE-extended models achieve significantly lower perplexity scores than those extended with YaRN, outperforming it by 0.081 on Llama2-7B and 0.137 on OLMo-7B. The difference between AlphaRoPE and NTK is less pronounced, which can be attributed to the fact that both methods, unlike YaRN, apply interpolation to high-frequency RoPE dimensions. This finding further supports our hypothesis that applying appropriate interpolation to high-frequency RoPE dimensions is beneficial for the capability of long-context modeling.

### 4.2.2 Passkey Retrieval

| Model Name | 4k | 8k | 16k | 32k | 48k | 64k | Overall |
|---|---|---|---|---|---|---|---|
| Llama2-NTK ($s = 16$) | 80.0 | 80.0 | 80.0 | 80.0 | 74.0 | 10.0 | 73.0 |
| Llama2-YaRN ($s = 16$) | **100.0** | **100.0** | **100.0** | **100.0** | **94.0** | 4.0 | 89.5 |
| Llama2-AlphaRoPE ($s = 16$) | **100.0** | **100.0** | **100.0** | **100.0** | 90.0 | **74.0** | **93.5** |
| OLMo-NTK ($s = 32$) | **100.0** | **100.0** | 98.0 | 86.0 | 76.0 | 56.0 | 81.0 |
| OLMo-YaRN ($s = 32$) | **100.0** | **100.0** | **100.0** | 92.0 | **90.0** | 60.0 | 88.5 |
| OLMo-AlphaRoPE ($s = 32$) | **100.0** | **100.0** | **100.0** | 98.0 | **90.0** | **76.0** | **93.5** |

Table 3: Passkey retrieval accuracy results under different RoPE scaling methods (%). The "Overall" column represents the average accuracy across all tested context lengths.

We evaluated model performance on the passkey retrieval task across context lengths ranging from 0 to 64k, using 4k length intervals. Table 3 shows the passkey retrieval accuracies under different RoPE scaling methods. The results demonstrate that AlphaRoPE significantly outperforms both YaRN and NTK across both model architectures, indicating that appropriate interpolation of high-frequency dimensions effectively enhances the model's ability to capture long-range information. The NTK method shows the weakest performance, suggesting that its introduced interpolation magnitude is too aggressive, requiring more extensive fine-tuning to be effective. We show detailed results in Appendix F.

### 4.2.3 LongBench Tasks

| Model | Single-Doc QA | | Multi-Doc QA | | Few-Shot | | |
|---|---|---|---|---|---|---|---|
| | Qasper | Multifieldqa | HotpotQA | 2WikiMQA | Trec | SAMsum | TriviaQA |
| Llama2-NTK ($s = 16$) | 4.59 | 5.50 | 1.88 | 2.54 | **65.00** | 38.89 | **89.66** |
| Llama2-YaRN ($s = 16$) | **5.60** | 5.33 | **2.68** | **2.85** | 62.33 | 38.87 | 88.08 |
| Llama2-AlphaRoPE ($s = 16$) | 5.22 | **6.49** | 2.19 | 1.94 | **65.00** | **39.76** | 88.63 |
| OLMo-NTK ($s = 32$) | 7.69 | **17.25** | **9.86** | **10.48** | 64.33 | 5.52 | **83.30** |
| OLMo-YaRN ($s = 32$) | 7.90 | 16.87 | 8.89 | 9.29 | 60.00 | 3.63 | 81.46 |
| OLMo-AlphaRoPE ($s = 32$) | **7.97** | 15.54 | 9.01 | 9.74 | **66.67** | **7.06** | 82.92 |

| Model | Summarization | | Code | | Synthetic | | Average |
|---|---|---|---|---|---|---|---|
| | GovReport | MultiNews | LCC | Repobench-P | PassageRetrieval | PassageCount | |
| Llama2-NTK ($s = 16$) | 13.95 | 4.65 | 66.49 | 52.62 | 0.00 | 0.00 | 26.60 |
| Llama2-YaRN ($s = 16$) | 14.20 | 6.16 | 65.24 | **53.29** | **0.50** | **0.22** | 26.57 |
| Llama2-AlphaRoPE ($s = 16$) | **20.14** | **9.52** | **66.75** | 52.37 | 0.11 | 0.00 | **27.55** |
| OLMo-NTK ($s = 32$) | 27.90 | 18.30 | 51.54 | **45.87** | 4.72 | 3.75 | 26.96 |
| OLMo-YaRN ($s = 32$) | 24.67 | 16.97 | 46.28 | 40.26 | **6.62** | 3.42 | 25.10 |
| OLMo-AlphaRoPE ($s = 32$) | **28.70** | **19.84** | **51.58** | 44.30 | 3.90 | **6.58** | **27.22** |

Table 4: LongBench-E performance comparison under different RoPE-scaling methods

We evaluate extended models using LongBench-E, a curated subset of LongBench containing 13 diverse tasks supported by comprehensive experimental data. As shown in Table 4, AlphaRoPE demonstrates superior overall performance compared to existing RoPE scaling methods. The advantage is particularly pronounced against YaRN, with improvements of 0.98 and 2.12 average points on Llama2-7B and OLMo-7B, respectively, indicating AlphaRoPE's more effective handling of long-context scenarios.

While AlphaRoPE's margin over NTK scaling is narrower (0.95 and 0.26 points on the two architectures), its performance proves substantially more stable. This stability is most evident in the Passkey Retrieval task, where NTK suffers a dramatic performance collapse while AlphaRoPE maintains strong capability. The contrasting results reveal a critical limitation of existing methods: YaRN's avoidance of high-frequency interpolation leads to consistent underperformance, while NTK's uniform scaling approach results in unpredictable failures on specific tasks. In comparison, AlphaRoPE achieves an optimal balance, delivering both competitive performance and reliable stability across diverse evaluation scenarios.

## 5 Conclusion

In this paper, we have shown that AlphaRoPE outperforms mainstream length extrapolation methods, including NTK and YaRN. As an explicitly defined method that introduces no training parameters, AlphaRoPE requires minimal implementation cost while delivering superior performance. Extensive evaluations demonstrate that fine-tuned models equipped with AlphaRoPE achieve state-of-the-art results across multiple long-context benchmarks while consistently maintaining their original short-context capabilities.

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

# A    Reproducibility

Our work prioritizes reproducibility, and a comprehensive effort has been made to ensure our results can be verified. The core methodology, including the mathematical derivation of our RoPE scaling function and the proposed high-frequency scaling amplitude metric, is detailed in Section 3. All experimental setups, including dataset preparation, model architectures, and training hyperparameters, are described in Section 4. We are committed to making our research accessible and plan to release the code and model checkpoints upon paper acceptance to facilitate full reproducibility and encourage future research.

# B    The Use of Large Language Models

Our paper utilized AI-assisted writing tools during the drafting and polishing stages to enhance linguistic fluency and grammatical accuracy. These tools helped refine the technical presentation while ensuring the integrity of our original research content and scientific findings.

## C    Derivation of the Coefficient 0.6

The coefficient value of 0.6 in the logarithmic form $\alpha = 0.6 \ln s$ was determined through empirical optimization on the Llama2-7B model with a scaling factor of $s = 8$. We conducted systematic experiments testing $\alpha$ values of 1.0, 1.25, and 1.5, with results indicating that $\alpha = 1.25$ yielded optimal performance.

Solving for the coefficient $a$ in the equation $\alpha = a \ln s$ with $s = 8$ and $\alpha = 1.25$:

$$a = \frac{\alpha}{\ln s} = \frac{1.25}{\ln 8} \approx 0.60 \tag{15}$$

## D    Details of A Value Calculation of YaRN

Following the settings and notations in (Peng et al., 2023) where $\alpha = 1, \beta = 32$, we can calculate two critical dimensions, $d_{alpha} = 90$ and $d_{beta} = 36$, corresponding to dimension index 45 and 18 respectively. We then obtain the $A$ value for the YaRN method on the Llama2-7B model as:

$$A = \big( \prod_{i=18}^{45} \big( \frac{1 - \gamma(r(i))}{s} + \gamma(r(i)) \big) \big)^{\frac{1}{45}},$$

$$\text{where} \quad r(i) = \frac{4096}{2\pi * 10000^{2i/128}}, \gamma(r(i)) = \frac{r(i) - \alpha}{\beta - \alpha} = \frac{r(i) - 1}{31} \tag{16}$$

## E    Short Tasks Evalutaion

| Model | Arc-C | TrQA | Hella | GSM8K | Average |
|---|---|---|---|---|---|
| Llama2-NTK (s=16) | **48.63** | 38.08 | 56.58 | 9.70 | 38.41 |
| Llama2-YaRN (s=16) | 47.56 | 38.89 | 57.01 | **11.14** | **38.65** |
| Llama2-AlphaRoPE (s=16) | 47.78 | **38.96** | **57.21** | 9.70 | 38.10 |
| OLMo-NTK (s=16) | 41.72 | 33.36 | 55.79 | 2.92 | 31.41 |
| OLMo-YaRN (s=16) | 41.47 | 34.56 | **56.15** | **3.11** | 33.45 |
| OLMo-AlphaRoPE (s=16) | 41.47 | **36.36** | 56.03 | 2.43 | **34.07** |

Table 5: Standard short tasks performance over extended Llama2-7B and OLMo-7B

Table 5 compares different RoPE extension methods on standard short-context tasks including Arc-Challenge(Arc-C) (Clark et al., 2018), TruthfulQA(TrQA) (Lin et al., 2021), Hellaswag(Hella) (Zellers et al., 2019), and GSM8k (Shi et al., 2022). The results show that AlphaRoPE maintains comparable performance to YaRN, while NTK exhibits slightly degraded capabilities across most evaluation metrics. This suggests that while all methods preserve fundamental short-context understanding, NTK's aggressive interpolation strategy may partially impair the model's ability to capture local contextual information. The minimal performance differences indicate that AlphaRoPE achieves an optimal balance between maintaining short-context performance and enabling long-context extension.

## F    Details of Passkey Retrieval

For each target length, we tested the retrieval capability at different relative depths where the passkey could be located. Specifically, we examined depth values spanning from 0 to 1.0 in increments of 0.1 (i.e., [0, 0.1, 0.2, ..., 0.9, 1.0]). At each depth position, we conducted 5 independent test cases to ensure statistical reliability of the results. The experimental results are presented below.

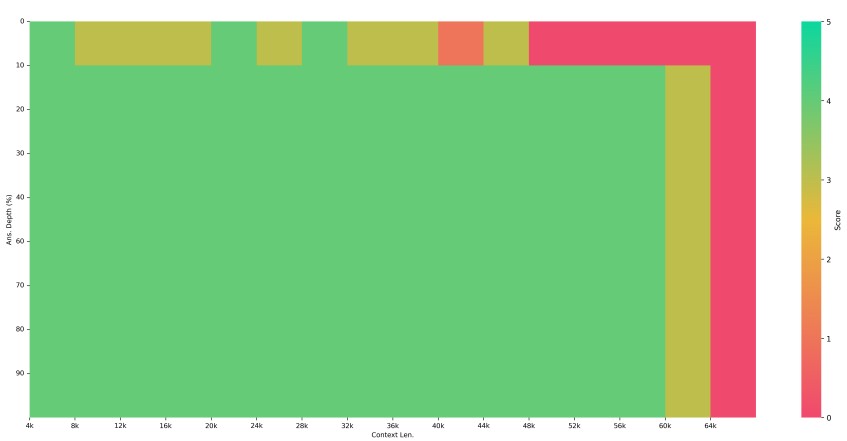

Figure 3: Passkey retrival result on Llama2-NTK (s=16)

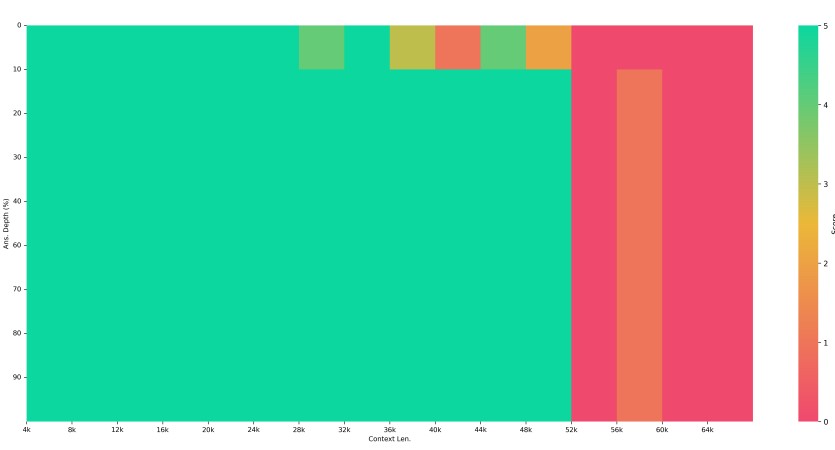

Figure 4: Passkey retrival result on Llama2-YaRN (s=16)

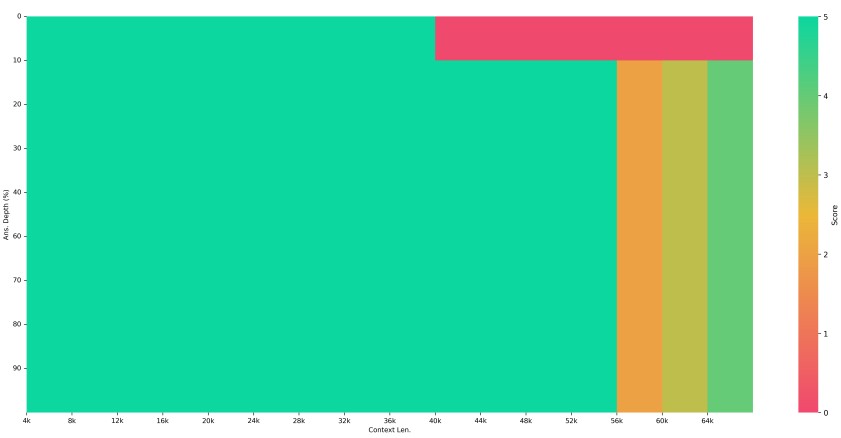

Figure 5: Passkey retrival result on Llama2-AlphaRoPE (s=16)

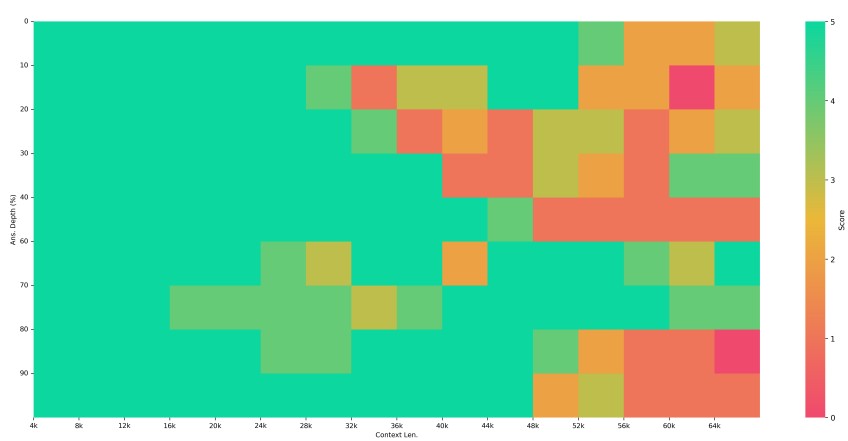

Figure 6: Passkey retrieval result on OLMo-NTK (s=32)

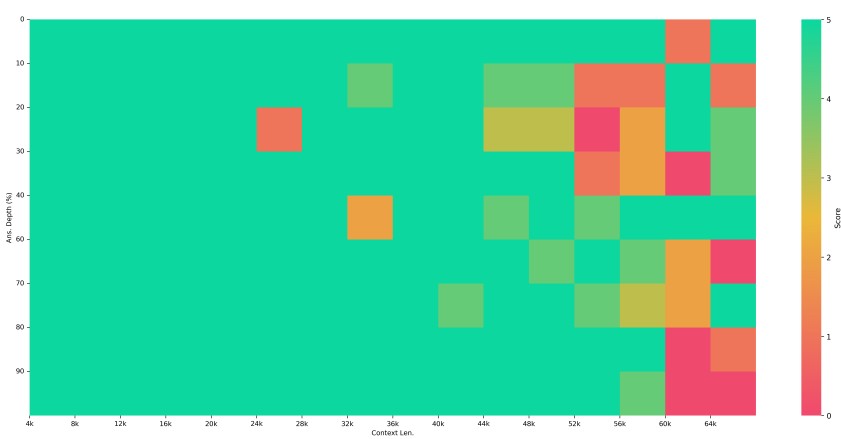

Figure 7: Passkey retrieval result on OLMo-YaRN (s=32)

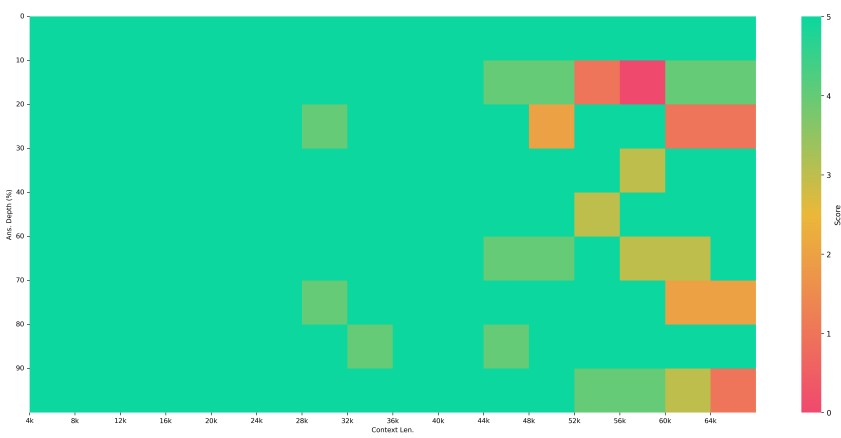

Figure 8: Passkey retrieval result on OLMo-AlphaRoPE (s=32)

