# OpenReview forum: "AlphaRoPE: A Simple Yet Effective Length Extrapolation Method"
_ICLR.cc/2026/Conference — ICLR 2026 Conference Withdrawn Submission_

### Official Review · Reviewer_ro2m · 2025-10-18

**Soundness:** 2
**Presentation:** 3
**Contribution:** 3
**Rating:** 6
**Confidence:** 4

**Summary:**

This paper introduces AlphaRoPE, a controlled, small-amplitude interpolation of highfrequency dimensions can better adapt RoPE-based LLM to longer contexts without compromising its ability to capture local context. This paper raises a critical issue with existing methods that their scaling factors on non-OOD dimensions are excessively large, leading to unnecessary positional disturbances.

AlphaRoPE improves upon existing methods by applying interpolation to the low-frequency components of the RoPE and a calibrated, gradually increasing interpolation to the high-frequency components. This approach balances capturing long-range dependencies with handling OOD issues, leading to better performance on long-context perplexity, Passkey Retrieval, LongBench and NIAH evaluation compared to traditional methods.

**Strengths:**

1. The scaling principles proposed for RoPE-based methods are cool and insightful.
2. The authors demonstrate a deep understanding of RoPE-based context extension and, crucially, take the innovative step of theoretical metric quantification for different interpolation methods.
3. The effectiveness of AlphaRoPE is verified on fine-tuned Llama2-7B and OLMo-7B in several standard long-context evaluation tasks and benchmarks.

**Weaknesses:**

1. The baseline models in this paper are rather outdated. Adding experiments on more recent LLMs such as Llama-3-8B, Llama-3.1-8B, and Qwen3-8B is necessary.
2. The evaluation lacks challenging long-context benchmarks, such as synthetic ones like RULER[1] or BABILong[3], or realistic tasks related to agents, multi-turn dialogue, or long generation, to demonstrate that AlphaRoPE is stronger enough than the widely used extrapolation methods.
3. In Figure 2, the RoPE scaling factor in different dimensions for NTK should not remain stable after the critical dimension. Instead, it should keep growing exponentially.
4. Although the authors claim that AlphaRoPE applies minimal interpolation to the highest-frequency dimensions, its A value is still larger than that of YaRN. This is confusing and raises the question of whether YaRN better satisfies the motivation of this paper, or whether the current version of metric quantification is still inadequate.
5. The paper lacks an in-depth discussion. For example, while the authors mention that they have conducted systematic experiments testing $\alpha$ values of 1.0, 1.25, and 1.5, no ablation results are shown. Providing these results and a deeper analysis would significantly strengthen the soundness of AlphaRoPE.
6. Have the authors considered comparing AlphaRoPE with non-RoPE-scaling-based context extension methods such as DCA or Self-Extend?

[1] RULER: What's the Real Context Size of Your Long-Context Language Models?. arXiv preprint arXiv:2404.06654 (2024).

[2] BABILong: Testing the Limits of LLMs with Long Context Reasoning-in-a-Haystack. Advances in Neural Information Processing Systems 37 (2024): 106519-106554.

[3] Training-free Long-Context Scaling of Large Language Models. arXiv preprint arXiv:2402.17463 (2024).

[4] LLM Maybe LongLM: Self-Extend LLM Context Window without Tuning. arXiv preprint arXiv:2401.01325 (2024).

**Questions:**

See Weakness.

---

### Official Review · Reviewer_mN6P · 2025-10-29

**Soundness:** 2
**Presentation:** 2
**Contribution:** 2
**Rating:** 2
**Confidence:** 3

**Summary:**

This paper introduces AlphaRoPE, a scaling strategy for Rotary Position Embeddings (RoPE) designed to extend the context window of Large Language Models. The core idea is to challenge the "high-frequency extrapolation" paradigm of existing methods like YaRN. The authors argue that, due to trigonometric periodicity, high-frequency dimensions can and should be interpolated (albeit gently) to enhance long-range dependency modeling. AlphaRoPE proposes a frequency-adaptive scaling function. Empirically, the authors conduct experiments on Llama2-7B and OLMo-7B. AlphaRoPE outperforms both NTK and YaRN on perplexity and passkey retrieval, while preserving short-context performance.

**Strengths:**

1. The paper's strongest motivation is the re-examination of high-frequency components in RoPE. The authors argue that "high-frequency extrapolation" (as used in YaRN) is an oversimplification and a controlled interpolation can leverage periodicity to capture long-range information.
2. The method works well on perplexity and passkey retrieval.

**Weaknesses:**

1. The paper's central claims of a "mathematically-derived" and "theoretically-derived" method are not supported by the evidence. Appendix C states: "The coefficient value of 0.6... was determined through **empirical** optimization..".
2. Because the core 0.6 coefficient is empirically tuned on one model (Llama2-7B) at one scale (s=8), there is no theoretical reason to believe this formula is optimal for other architectures (e.g., Mistral, Llama 3) or, more importantly, for other model scales (e.g., 70B, 175B). The paper claims generalizability but has only demonstrated it on one other 7B model. The strong theoretical framing is used to justify this generalization, but since the theory is just an empirical fit, the justification dissolves.

**Questions:**

Given that the 0.6 coefficient is empirical, could the authors re-frame the paper to be more transparent about this? The contribution would still be valuable as a new, effective empirical method, but the current theoretical claims are misleading.

---

### Official Review · Reviewer_DLmu · 2025-10-30

**Soundness:** 3
**Presentation:** 3
**Contribution:** 3
**Rating:** 4
**Confidence:** 4

**Summary:**

This paper proposes AlphaRoPE, a new method for extending the context length of Large Language Models using Rotary Position Embeddings (RoPE). The method challenges the conventional wisdom of treating high-frequency RoPE dimensions as purely local, arguing that their periodic nature allows them to contribute to long-range dependencies. AlphaRoPE introduces a novel scaling strategy that applies a controlled, gradually increasing interpolation to high-frequency components while using standard interpolation for low-frequency ones. This is achieved through a mathematically derived power function that aims to balance local context preservation with long-range extrapolation. Experiments on Llama2-7B and OLMo-7B models show that AlphaRoPE outperforms existing methods like NTK and YaRN on perplexity, passkey retrieval, and LongBench-E benchmarks.

**Strengths:**

1. The paper is well-structured, providing a clear theoretical motivation for its design choices, which are then validated through a comprehensive set of experiments.

2. The central insight that high-frequency components can contribute to long-range dependencies via trigonometric periodicity is compelling and provides a solid justification for moving beyond the strict extrapolation/interpolation dichotomy of prior work.

3. AlphaRoPE is presented as an explicit, parameter-free (post-derivation) scaling function, making it easy to implement and apply without the need for expensive evolutionary searches or additional training parameters.

**Weaknesses:**

1. While the paper frames the interpolation of high frequencies as a fundamental reconsideration of RoPE scaling, the empirical results suggest its impact is more of an incremental refinement than a paradigm shift. The overall architecture still relies heavily on the established principle of aggressive interpolation for low frequencies (j > d₀) and minimal changes for high frequencies. This makes the contribution feel more like a well-engineered trick to smooth the scaling curve and eke out extra performance, rather than a completely new approach to length extrapolation.

2. The performance gains, while consistent, are modest on the more realistic and diverse LongBench-E benchmark, especially for the Llama2-7B model. While the method shows clear superiority on the synthetic Passkey Retrieval task, its advantage shrinks significantly on complex, real-world tasks. This raises questions about the practical significance of the proposed changes.

3. The paper lacks a sensitivity analysis or ablation study on the critical dimension $d_0$, which marks the boundary between the high and low-frequency scaling regimes. The entire method hinges on this threshold, yet its value is adopted from prior work without investigation. An analysis of how performance changes with different choices of $d_0$ would be crucial for understanding the robustness of AlphaRoPE and whether its effectiveness is highly dependent on this specific hyperparameter.

**Questions:**

1. The coefficient 0.6 in the formula for $\alpha$ was determined empirically on Llama2-7B. How does this coefficient generalize to other model architectures and sizes? Have you tested whether this value needs to be re-tuned for different models to achieve optimal performance?

---

### Official Review · Reviewer_j9vH · 2025-11-03

**Soundness:** 3
**Presentation:** 4
**Contribution:** 2
**Rating:** 4
**Confidence:** 4

**Summary:**

The paper claims that appropriate interpolation of high-frequency RoPE dimensions can enhance their contribution to long-range dependency encoding through trigonometric periodicity. The authors extended the RoPE model by a scaling strategy that achieves optimal balance between local preservation and long-context adaptation through controlled high frequency interpolation.

**Strengths:**

The paper is easy to follow and well-written.
Figure 1 and texts are provided to demonstrate the underling motivation of the proposed model.
Experiments were conducted with a number of LLMs.

**Weaknesses:**

See the below questions.

**Questions:**

It is claimed in the introduction section that “high-frequency dimensions can also contribute to encoding long-range dependencies”. How many precents of such high-frequency dimension needed to encode long-range dependencies? Are the any examples/figures that can further illustrate this motivation?

This is an incremental work that extends from the RoPE algorithm and other RoPE-based algorithms, resulting in the fact that the novelty and contribution of the paper is marginal and not significant.

Related work sections should be updated to include most recent work that published in 2025, for instance.

Other RoPE-based algorithms should be taken into account as baselines in the experiments for comparisons.

Are there other datasets and tasks to be included in the experiments?

---

### Note · Authors · 2025-11-22

I have read and agree with the venue's withdrawal policy on behalf of myself and my co-authors.